# Preharvest Sprouting in Quinoa: A New Screening Method Adapted to Panicles and GWAS Components

**DOI:** 10.3390/plants13101297

**Published:** 2024-05-08

**Authors:** Cristina Ocaña-Gallegos, Meijing Liang, Emma McGinty, Zhiwu Zhang, Kevin M. Murphy, Amber L. Hauvermale

**Affiliations:** Department of Crop and Soil Sciences, Washington State University, Pullman, WA 99163, USA; c.ocanagallegos@wsu.edu (C.O.-G.); meijing.liang@wsu.edu (M.L.); emma.mcginty@wsu.edu (E.M.); zhiwu.zhang@wsu.edu (Z.Z.)

**Keywords:** PHS screening, panicle-wetting test, seed dormancy, genome-wide association study (GWAS)

## Abstract

The introduction of quinoa into new growing regions and environments is of interest to farmers, consumers, and stakeholders around the world. Many plant breeding programs have already started to adapt quinoa to the environmental and agronomic conditions of their local fields. Formal quinoa breeding efforts in Washington State started in 2010, led by Professor Kevin Murphy out of Washington State University. Preharvest sprouting appeared as the primary obstacle to increased production in the coastal regions of the Pacific Northwest. Preharvest sprouting (PHS) is the undesirable sprouting of seeds that occurs before harvest, is triggered by rain or humid conditions, and is responsible for yield losses and lower nutrition in cereal grains. PHS has been extensively studied in wheat, barley, and rice, but there are limited reports for quinoa, partly because it has only recently emerged as a problem. This study aimed to better understand PHS in quinoa by adapting a PHS screening method commonly used in cereals. This involved carrying out panicle-wetting tests and developing a scoring scale specific for panicles to quantify sprouting. Assessment of the trait was performed in a diversity panel (*N* = 336), and the resulting phenotypes were used to create PHS tolerance rankings and undertake a GWAS analysis (*n* = 279). Our findings indicate that PHS occurred at varying degrees across a subset of the quinoa germplasm tested and that it is possible to access PHS tolerance from natural sources. Ultimately, these genotypes can be used as parental lines in future breeding programs aiming to incorporate tolerance to PHS.

## 1. Introduction

Preharvest sprouting (PHS) occurs in seeds with little or no dormancy and results in seeds sprouting on the spike after a rain event and prior to harvest [1,2,3]. PHS causes losses in yield and seed viability and negatively impacts end-use quality [4]. It also limits the expansion of quinoa (*Chenopodium quinoa*) to humid regions and might become increasingly prevalent under climate change conditions affecting weather patterns in fields around the world [5,6,7]. PHS has been extensively studied in major cereal crops, such as wheat and barley [2,8,9,10,11,12], but not in underutilized crops like quinoa. There is little information and research about PHS in quinoa because it has only recently emerged as an agronomically important crop worldwide, and differences in its plant architecture compared to that of cereals make the use of conventional PHS screening methods more challenging.

Literature on quinoa PHS is limited and fails to report quantitative data on yield loss or decreased seed quality. The first report of this trait in quinoa came from Ceccato et al. [4], who found two genotypes, Chadmo and 2-want, exhibiting dormancy at harvest under field conditions. Additional dormancy testing using plating assays helped to confirm the presence of seed dormancy in both genotypes and indicated they were potential sources of PHS tolerance. Ceccato et al. [12] shed light on the factors affecting seed dormancy and germination, which included temperature, water stress, salinity, hormonal control, aging, and storage. Bertero and Benech-Arnold [13], as cited in Ceccato et al. [12], focused on a small panel of quinoa genotypes (15 to 20) and noted that most of those quinoa cultivars lacked dormancy and hence were highly susceptible to PHS in field conditions.

To date, McGinty et al. [7,14] have provided the most complete theoretical framework for quinoa seed dormancy. Seed dormancy is an evolutionary adaptation that prevents seeds from germinating to survive natural catastrophes and ensures species survival [15] Primary dormancy includes physiological, morphological, and physical dormancy and is established during development, whereas secondary dormancy is established by environmental factors after primary dormancy is lost [16,17]. In terms of hormone signaling, if quinoa behaves like cereal behaves, like other cereal seeds, then dormancy is also regulated by ABA and GA [14,18,19,20,21]. Studies in quinoa, and relatives of quinoa, such as *C. album* and *C. berlandieri*, indicate the presence of primary dormancy in some genotypes. In others, there appears to be an absence of dormancy. Collectively, this suggests that different quinoa genotypes have mixed dormancy types, i.e., primary dormancy is present in some genotypes and absent in others [7,14]. Quinoa dormancy studies provide evidence of a wide range of seed dormancy strength across the diverse germplasm, which may have implications for PHS tolerance or susceptibility [14].

Assessment of PHS in cereals is routine, employing three methods: visual observation, Hagberg–Perten falling numbers (FNs), and alpha-amylase enzyme assays. Visual evaluation of sprouted shoots and roots is based on scales specific to inflorescence types, such as spikes, or based on increases in the inflorescence area [3,22]. The spike-wetting test is inexpensive to perform and easy to use on breeding populations. FN assessment consists of mixing wholemeal flour with water, heating it up, and letting a stirring paddle fall through the solution; flours from PHS grains are less viscous and allow the paddle to fall faster, producing undesirable low FN numbers ([23], FGIS Directive 9180.3, 2019). Lastly, elevated activity of alpha-amylase, an enzyme that catalyzes the hydrolysis of starch and negatively affects the end-use quality of cereal products, can be indicative of PHS. Measurement of alpha-amylase activity can occur indirectly using FNs or using highly specific assays [24,25]. Unlike the routinely used methodology for PHS testing in cereals, only the use of alpha-amylase enzyme assays has been previously reported in quinoa, though in the context of end-use quality for baking or nutrition, not in the context of PHS [26,27,28]. Moreover, the FN method and alpha-amylase enzyme assays are costly to perform compared with a spike-wetting test [25]. Overall, these methods have not been validated in quinoa but could be adapted to the crop for PHS assessment. In fact, this study aims to translate the visual evaluation performed on wheat spikes to the panicles of quinoa, given that the FN method and alpha-amylase enzyme assays are costly to perform compared to wetting tests [25].

It is also common to use germination tests at physiological maturity as a proxy for PHS tolerance or susceptibility given the association between a lack of seed dormancy and PHS prevalence in cereals [3,16]. Germination tests using threshed seeds at physiological maturity provide an overview of seed dormancy status but do not consider the morphological components of the inflorescence that may increase or decrease PHS susceptibility. For example, in wheat, Paterson et al. [8] described the erectness of spikes, the openness of florets, and the level of germination-inhibiting compounds in the bracts as important traits that influence seed sprouting in wetted spikes. Also, wider awn angles and longer awns maximize water exposure in the ear and are associated with a larger incidence of PHS [29,30]. In quinoa, the panicle shape, density, leafiness, and other features may impact PHS. However, there is no report of a PHS panicle-wetting screen for quinoa in the literature.

To test the hypothesis that some quinoa genotypes have a certain degree of PHS tolerance, panicle-wetting tests were developed. Wetting tests considering the plant’s morphological features were carried out on a diverse panel of quinoa (N = 336), representing a subset of the World Core Collection (WCC). The screening methods used here constitute a modified version of the conventional PHS screening methods used in wheat [8]. A novel PHS scoring system for quinoa based on the relative percentage of sprouted seeds per panicle was developed with the objective of quantifying the trait in wetted panicles. Genotype rankings were created based on two metrics: the sprouting index (SI) and rate of sprouting (slope values). The SI is a weighted index that gives higher scores to early sprouting plants and progressively less weight to those sprouting later [31,32]. The current study aimed to identify natural sources of PHS tolerance in the quinoa genetic pool that could serve as parental lines in future quinoa breeding programs.

The collected phenotypes were used with genotypic data from a subset of genotypes to carry out a genome-wide association study (GWAS). GWAS research on PHS is vast in cereals like wheat and barley but very limited in underutilized crops like quinoa. The current literature on PHS in quinoa is limited to one study by Lopez-Marques et al. [33], who, based on a phylogenetic analysis, determined MFT (Mother of Flowering Time and TFL1) and MKK3 (mitogen-activated protein kinase 3) to be homologous genes involved in PHS regulation. If the nature of PHS in quinoa similar to that of cereals, the GWAS in this study would be expected to identify similar genes, or chromosomal regions, underpinning the trait. In wheat alone, a total of 110 quantitative trait loci (QTLs) linked to PHS resistance have been reported, and these are often related to the molecular signaling processes of the phytohormones abscisic acid (ABA) and gibberellin (GA) [21,34]. The current study used the Genome Association and Prediction Integrated Tool (GAPIT v3) R package [35], incorporating the BLINK (Bayesian-information and Linkage-disequilibrium Iteratively Nested Keyway) method [36], to identify single-nucleotide polymorphisms (SNPs) associated with PHS in quinoa. GAPIT v3 is a robust tool for genomic association analysis. Compared to previous versions, it incorporates multi-locus test methods such as BLINK, which significantly enhance its statistical power for GWAS and reduce the computing time for analyzing large genomic datasets.

## 2. Results

The PHS phenotype followed a geometrical distribution, meaning that the score values started low and only increased over time until a maximum value was reached. This was evidenced as a proportion of higher scores increasing over time (Figure 1).

However, the spread of scores on planting date 2 (PD2) was larger than on planting date 1 (PD1), likely due to delays in harvesting on PD2 attributable to the synchronous maturity of all the genotypes (Figure 2).

Given the substantial number of accessions in this study, significant differences were only demonstrated for five of the top and bottom genotypes from the PHS tolerance rankings. SI plotting of these 10 genotypes against the two controls showed significant differences, as shown by the standard error bars (Figure 3).

In the panicle-wetting tests (duration = 7 days), most of the genotypes had average scores ranging between 0 and 1 on day 1. The overall mean sprouting score was 0.58 on day 1 and 1.76 on day 2, whereas by day 3, this score almost doubled to 3.31. Halfway through the panicle-wetting tests, at day 3, both extremes of the sprouting spectrum were observed, with some genotypes showing average sprouting scores lower than 1 (genotypes 37, 221, 164, 287, 148, 181, 179, 140, 169, 162, 144, 50, and 300) and others showing average sprouting scores equal or greater than 7 (genotypes 316, 315, 312, 304, 281, and 108; Appendix A). By day 5, 70% of the panel had scores equal or greater than 5, which translates to having relative sprouting of 40% or more after 5 days of misting.

### 2.1. Genotype Rankings for PHS Tolerance

A simple linear regression model was fitted, and XY plots were used to reflect the increase in the sprouting scores over seven days. The genotypes with the most PHS tolerance displayed slower increases in slope, and the genotypes with the greatest PHS sensitivity showed the steepest slopes (Appendix A). A ranking based on ascending slope values for the panel is proposed in Appendix A. Here, 16 genotypes showed lower scores than the PHS-tolerant control “Redhead”(ID #300). These 16 included 46, 221, 181, 164, 142, 344, 224, 179, 287, 267, 192, 37, 169, 290, 144, and 162. The genotypes displaying the most sprouting had PHS scores between 1 and 3 or 5 in more extreme cases on day 1 and reached the maximum value of 9 before day 7. The genotypes falling into this group included 31, 108, 134, 208, 281, 292, 295, 320, 327, and 357. The PHS-susceptible control “QQ74” (ID #9) ranked 309 out of 334.

The daily PHS averages per genotype were also used to calculate the individual SIs, which ranged from 0.190 to 2.530, where low values are reflective of more PHS tolerance and large values reflective of PHS susceptibility (Appendix A). The top 12 genotypes with the lowest SIs were 221, 37, 181, 164, 287, 142, 179, 46, 163, 144, 190, and 169, all lower than the PHS-tolerant control “Redhead” (ID #300). The genotypes with the highest SIs included 315, 312, 70, 281, 316, 320, 330, 304, 108, and 227 and were all below the PHS-susceptible control “QQ74” (ID #9), which ranked 262 out of 334. The Pearson’s correlation value between slope and SI is 0.782.

### 2.2. Addressing Population Stratification through PCA and an Indicator

The first three principal components (PCs) collectively accounted for more than 50% of the observed phenotype variance. The first PC elucidated 36.7% of the variance within the phenotype, indicating its substantial contribution to the overall variance. Following PC1, the second PC explained 10.71% of the variation, while PC3 contributed 2.9% to the overall variance. Analysis of these PCs allowed us to identify a distinct population structure. The PC plots (Figure 4) illustrate the separation of the quinoa population into two clusters: one characterized by positive PC1 values and the other by negative PC1 values.

To further mitigate potential confounding effects, we constructed a Coefficient of Variation (CV) utilizing the first three PCs and an additional indicator. This composite measure helped to reduce false positive results during the association analysis (Appendix A).

### 2.3. Genome-Wide Association Study and Functional Annotations

Our investigation identified 19 significant markers associated with the pre-harvest sprout (PHS) trait in quinoa, located on chromosomes 2B, 4B, 5A, 6B, and 9B (Figure 5). Among these, five markers are related to the PHS upper asymptote (parameter a), with the maximum phenotypic variation explained by Cq9B_243431086 (*p* = 1.61 × 10^−9^, PVE = 76.8%). Additionally, six markers are associated with the PHS growth rate (parameter b), with Cq7A_17313829 explaining the maximum phenotypic variation (*p* = 1.97 × 10^−9^, PVE = 83.8%). Furthermore, eight markers are linked to PHS maximum growth (parameter c), with Cq6A_8709044 explaining most of the phenotypic variation (*p* = 2.57 × 10^−17^, PVE = 91%). Notably, the most significant marker, Cq2A_11615394 on chromosome 2A, is related to maximum growth. Seventeen markers had a matching locus on NCBI according to blasting with the *Chenopodium quinoa* genome as the reference. Functional annotation revealed five GO terms, including integral component of membrane, transcription factor activity, sequence-specific DNA binding, rRNA processing, and calcium-transporting ATPase activity, among others. These terms are related to KEGG pathways such as transporters, aminoacyl-tRNA biosynthesis, and metabolic pathways (Table 1).

## 3. Discussion

This study aimed to better understand PHS in quinoa by adapting a PHS screening method commonly used in cereals and by developing a scoring scale specific to panicles. To assess sprouting diversity, experiments were performed on a diversity panel comprising 336 quinoa genotypes.

Variability in PHS tolerance was expected given the genetic diversity of quinoa and previous reports of phenotypic variation among genotypes [37]. Variation in PHS was observed and partially matched the results from previous assessments. For example, Peterson and Murphy [38] reported observing less sprouting for genotype PI-614880 (PHS ID #223), which is consistent with the sprouting index and slope values reported in the current study. Genotype PI-614880 scored 16th out of 336 in the PHS tolerance ranking, with an SI = 0.627 and a slope value of 0.715. Another case was genotype PI-614880, which has several identifiers, including the names Chadmo, NSL-106393, and QQ065. Ceccato et al. [4] reported Chadmo, or PI-614880 in this study, as a potential source of PHS tolerance. The correlation between the slopes and SI values (0.751) is indicative of a linear relationship between the parameters and gives confidence for their use.

With the exception of four varieties (Ames-13725, Ratuqui, D-11924, and Ames-13740), this study evaluated sprouting in the same varieties previously screened for the presence of seed dormancy in [14]. It was initially hypothesized that lower sprouting scores would be associated with varieties identified with stronger seed dormancy. In fact, this was observed for genotypes CHEN-291, PI-614883, CHEN-299, and D-12020. However, it was also the case that 43 genotypes identified with some degree of seed dormancy in [14] showed susceptibility to sprouting in this study. Namely, both Redhead and QQ74, the genotypes used as the tolerant and susceptible controls for this study, were identified as having no seed dormancy at physiological maturity [14]. Taken together, these results suggest that seed dormancy is only one component of PHS tolerance in quinoa and that after-ripening time and panicle architecture are also important. The role of panicle architecture is supported by the observation that although both controls used in the current study were originally classified as non-dormant [14], they displayed vastly different degrees of resistance to sprouting. Both were selected as controls based on the observed presence or absence of sprouting in the field. Redhead, selected as the PHS-tolerant control, had an SI of 0.534, consistent with an increased PHS tolerance, whereas our susceptible control (QQ74) had an SI of 1.643, consistent with lower PHS tolerance. Future work will need to evaluate sprouting scores in parallel with seed dormancy screens to understand the impacts of after-ripening on PHS status. Additionally, the relationship between panicle morphology, the impacts of domestication, and PHS will need to be investigated.

Though this study provides quantitative data (SI and slope values) to determine the tendency of quinoa genotypes towards PHS susceptibility or resistance, it did not set a threshold to categorize the different levels in the spectrum. PHS is a quantitative trait; it is complex to divide observations into either resistant or susceptible given that PHS exists as a degree of tolerance in each of the quinoa genotypes. Studying the PHS trait over seven days added complexity to our results, as a given genotype can appear resistant to PHS on day 1 but become susceptible after 7 days of misting. Integration of the temporal data into the SI and slope values allowed us to look at each genotype holistically. Like Rasul et al. [39], future research on quinoa could propose well-defined levels of PHS resistance (susceptible, tolerant, and resistant) based on SI, FN, and germination index measurements.

The PHS scores were consistently low for the PHS-tolerant control (Redhead) and higher for the PHS-susceptible (QQ74) control across the batches and PDs. It is important to note that these control genotypes were chosen based on their observed percent sprouting in the field rather than PHS screening and did not end up representing the most tolerant or resistant lines found within the WCC. However, they still represented contrasting ends of the sprouting spectrum and served as a reference and a point of comparison for the rest of the genotypes in the ranking. Considering the slope and sprouting index values together, our rankings identified ten genotypes with less sprouting than Redhead (PHS-tolerant control) and could help future PHS research to select even more PHS-susceptible or PHS-tolerant genotypes than those used as controls in this study. Genotypes with phenotypic divergence, or in this case with contrasting PHS phenotypes, are valuable for plant breeding studies aiming to develop recombinant inbred lines (RILs) to fine-map the regions where causative loci lie and to develop markers for marker-assisted selection [34].

Planting date (PD) was included in the experimental design to provide a complete replication in time and space of the misting experiments. Differences between PD1 and PD2 were observed, with the PHS scores from PD2 being overall higher than those from PD1. An explanation for this is the delay in harvesting for the panicles from PD2, which were harvested over a longer period than PD1. Batches were included in the experimental design for logistical reasons; that is, to allow for handling of a smaller number of plants each week according to labor capacity. The sprouting results for the controls were consistent throughout the experiment; that is, Redhead sprouted less compared to QQ74. Though we believe that the effect of batch is negligible, possible sources of variability include differences in scoring from person to person and variation in scoring across time. These are all likely explanations for the lower-than-average scores in batch 2 and for the overall differences across batches (data not shown).

Our GWAS results identified 19 significant SNPs from the panicle-wetting assays. Given quinoa’s status as a relatively new crop with limited annotation information for downstream GWAS, we employed bioinformatic methods using BLASTing and the DAVID database to predict the potential functionality of the PHS-associated markers. Although the identified SNPs were not associated with previously identified PHS markers in wheat, the analysis showed that 10 of the identified SNPs may be associated with seed dormancy and germination (Table 1). More specifically, basic leucine zipper repeat (bZIP) proteins, mitochondrial transcription termination-factor (mTERF), and pentatricopeptide repeat (PPR) proteins play roles in osmotic stress responses and ABA signaling and sensitivity [40,41,42,43,44,45], whereas ABC transport G-family-like, G-type lectin S-receptor-like, Main-Like 1, and Peter Pan-like proteins function in seed development, germination embryogenesis, cell division, and elongation in many plants, including Arabidopsis, barley, maize, and rice [43,46,47,48,49]. The predicted results could provide valuable insights into future research directions and assist other researchers in making informed decisions regarding marker design or gene cloning in their breeding population. Future work will need to investigate the robustness of the identified SNPs as markers for improving PHS tolerance in quinoa.

## 4. Materials and Methods

### 4.1. Germplasm

A total of 315 accessions from the quinoa WCC, 19 entries from Washington State University’s quinoa variety trials, and 2 Ecuadorian varieties donated by Angel Murillo at INIAP Ecuador were used in the current study. The subset used is part of the WCC, an international collection of accessions assembled by researchers at KAUST to represent high levels of geographic and genetic diversity. All the information on the accessions used, including study number, accession name, and geographical region of origin, is provided in Appendix A.

### 4.2. Experimental Design and Greenhouse Conditions

A complete randomized design (CRD) was used for the panicle-wetting tests to ensure equal distribution of the environmental gradient inside the greenhouse across all the panicles. All genotypes had at least 3 replicates in total, and when available, four replicates were analyzed on each of the two planting dates (PDs). The number of biological replicates included for each genotype is shown in Appendix A. The second PD was separated from the first by three weeks to allow for complete experimental replication across time and space. Due to the vast genetic diversity in quinoa, physiological maturity occurs at different times after sowing. To accommodate differences in maturity date and to streamline screening, the genotypes were planted in a staggered manner. Long-maturing genotypes were planted first, and short-maturing genotypes were planted last to synchronize harvest at physiological maturity. Batches were included in the experimental design to accommodate the large number of plants. The experiment was broken down into 11 parts, starting with plants from PD1 and continuing to PD2. Each batch contained between 100 and 300 panicles. To negate any effects of the batch, control genotypes were included in every batch. Controls were selected based on previous field observations (Figure 6) and included 3 PHS-tolerant (Redhead genotype) and 3 PHS-susceptible (QQ74 genotype) plants per batch.

### 4.3. Development of the PHS Scoring Scale

Given the lack of literature and methods for assessing PHS in quinoa, a scoring scale based on the relative percentage of sprouted seeds was developed based on similar studies in wheat [8,50]. The initial step consisted of taking a reference picture of a sprouted panicle and dividing the image into clusters; sprouted seeds were counted individually and compared to the total number of seeds per cluster to obtain the average sprouting in the panicle (Appendix A). This process was developed into a visual scoring scale with 6 levels, each representing a range of the sprouting percentage in the panicle (Figure 7). The defined levels were 0 (0% sprouting), 1 (1–19% sprouting), 3 (20–39% sprouting), 5 (40–59% sprouting), 7 (60–79% sprouting), and 9 (>80% sprouting). This scoring system used odd numbers, similar to plant-level phenotyping for traits like panicle density [51,52], to maintain consistency with quinoa phenotyping methodologies.

### 4.4. Data Collection and Analysis

The total number of harvested plants, excluding the controls, was 2424, where 1267 corresponded to PD1 and 1157 to PD2. The sprouting scores were recorded daily for each panicle, using the PHS scale developed here. Data analysis was carried out with 2250 plants, corresponding to 336 genotypes (controls included), and excluded genotypes with poor seed development or that molded during misting. All the data were analyzed using R 4.1.2 software [53]. The packages used to visualize the distribution, discover patterns, and spot anomalies included tidyverse [54], naniar [55], visdat [56], and ggplot2 [57].

The average sprouting per day of all the replicates in each genotype was calculated using the recorded scores. The genotype averages per day were used for two calculations: slope (rate of sprouting over time) and sprouting index (SI), defined as (7 × Sday1 + 6 × Sday2 … + 1 × Sday7)/(7 × n), where S corresponds to the PHS score each day and n is the maximum sprouting score [45]. A starting value of 0 on day 0 was added to all the genotypes before fitting a simple linear model. Then, the slopes were plotted in XY plots with the R lattice package [58], and the slope values were extracted and organized in ascending order from the lowest to highest slopes to rank all the genotypes.

### 4.5. Phenotypic and Genotypic Data Integration for PHS Association Study

A subset of 279 quinoa accessions for which genotypic data were available was used for GWAS. The averages for each accession were calculated using Excel PivotTables. The calculation was based on the scores from seven days, with eight replications, and two different planting dates. These average PHS scores were then integrated into sigmoid curves using the equation:$$y\;=\;\frac{a}{1-e^{-b\left(t-c\right)}}$$
where *y* represents the average PHS score, *a* denotes the upper asymptote, *b* is the growth rate, *c* is the time of maximum growth, and *t* corresponds to the day on which the observation was made (1 to 7), following the approach outlined by McCulloch and Pitts [59].

The genotypic data, provided by Patiranage et al. [37], were meticulously processed. Sequences with less than five reads were excluded, and SNP markers with a missing rate exceeding 20% or a minor allele frequency below 5% were removed. A total of 48,025 markers were retained for the subsequent association analyses. The imputation procedures were executed using GAPIT v3 [35].

### 4.6. Population Structure Analysis

To address potential confounding effects arising from population stratification, a robust approach leveraging the first three principal components (PCs) and an indicator derived from the first PC, designated as the coefficient of variation (CV) matrix, was applied within our association model. Principal Component Analysis (PCA) was executed using GAPIT v3, utilizing the genotype data. A threshold of 0 for the PC1 values was employed as the criterion for identifying population stratification. Specifically, PC1 values greater than or equal to 0 were coded as 1, while those smaller than 0 were coded as 0. Visualization of the population structure was accomplished through PC plots generated using the R ggplot2 package [57]. This step ensured appropriate adjustment of population structure effects for the association study.

### 4.7. Association Study for Integrating Planting Date Effects Using an Additive–Additive (AA) Model

This study collected phenotype data across two distinct planting dates: PD1, comprising 229 accessions, and PD2, comprising 228 accessions. Notably, 211 accessions are shared between the two PDs, while each accession within a single PD exhibits unique phenotype data. To consolidate phenotypes, genetic effects, and PD information, we implemented a restructured additive–additive (AA) model for the association study (Figure 8). The original genotype matrix represents genotypes as 0, 1, and 2, with 0 and 2 indicating homozygous genotypes and 1 representing heterozygous genotypes. In the AA model, we recode the genotypes into 0 and 1, designating 0 for heterozygous genotypes and 1 for homozygous genotypes. Following the recoding process, the genotype matrix is diagonally and symmetrically adjusted before being concatenated vertically and horizontally with two zero matrices. This transformation prepares the data for the association analysis. The association study was executed using the BLINK method [30] implemented in GAPIT v3.

## 5. Conclusions

Overall, this research was successful at implementing PHS screening in quinoa and capturing the variability across genotypes. This initial screening assisted in the identification of natural sources of PHS tolerance within a diverse quinoa panel and constitutes the first step in the selection of elite parental lines for future breeding programs. The results presented here may support efforts to grow regionally adapted quinoa varieties, which will provide farmers with an alternative to diversify their cropping systems and eliminate the risk of yield loss.

Future experiments looking to replicate this study could choose not to stagger the planting date and solely rely on storing the panicles at −20 °C after harvest, which has been shown to preserve dormancy status in quinoa [14,60,61,62]. Our research aimed to have all the plants reach maturity by the same date to avoid differences in the post-harvest conditions, but it was logistically challenging to harvest all the plants at once.

Future research will look at defining mathematical thresholds for PHS susceptibility and tolerance based on the results provided in the current study. Additionally, measurements such as FNs or enzyme assays that measure alpha-amylase activity may complement our visual results and improve the current understanding of quinoa seed dormancy.

## Figures and Tables

**Figure 1 plants-13-01297-f001:**
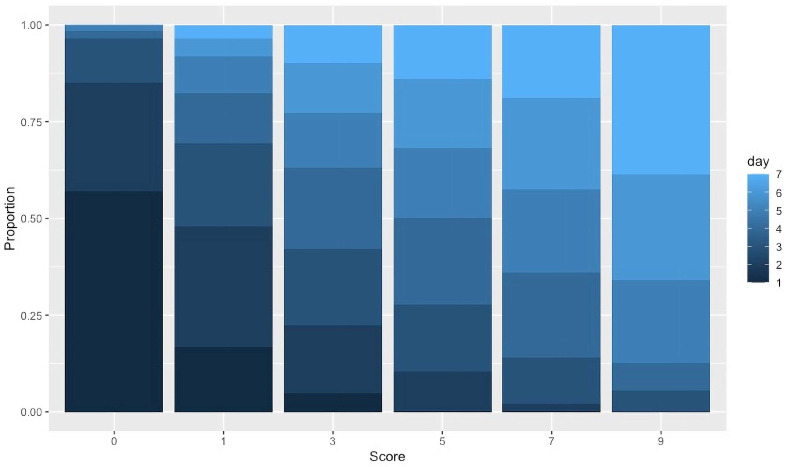
Proportion of observations by day for each PHS score level. PHS scores on the x-axis represent range of percent germination (levels 0 to 9), and the y-axis is the proportion of a level observed on a given day.

**Figure 2 plants-13-01297-f002:**
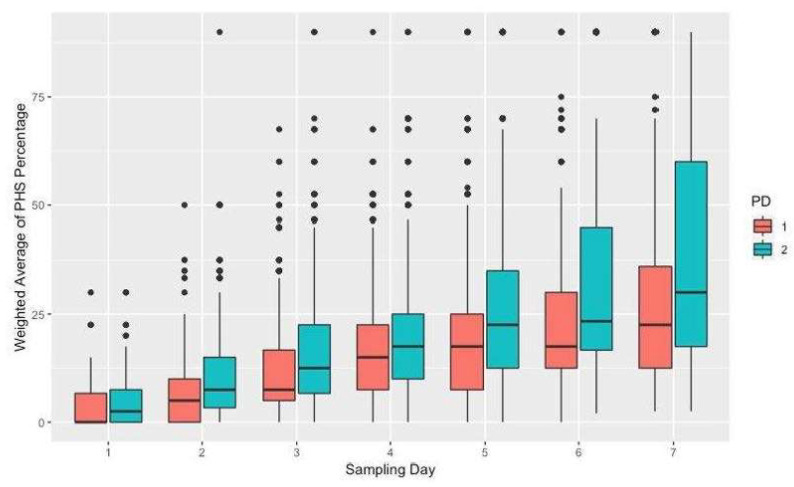
Distribution of PHS score averages by day and planting date (PD). PHS scores were multiplied by 10 to reflect sprouting category medians and are referred to as “weighted”. Outliers are indicated with black dots.

**Figure 3 plants-13-01297-f003:**
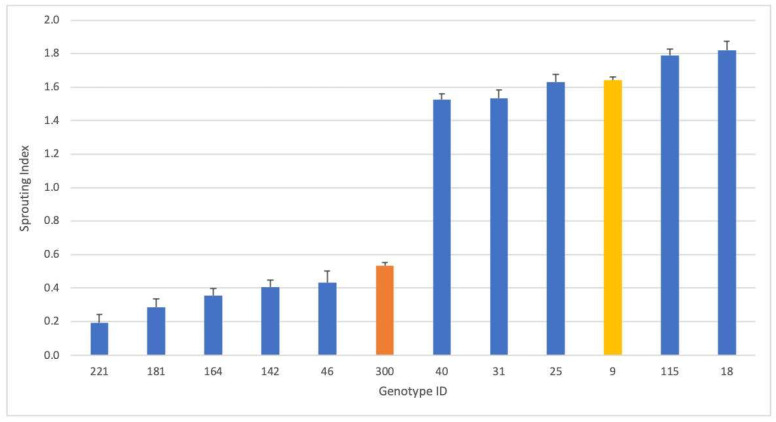
Sprouting index of 10 contrasting quinoa genotypes (blue), plus 2 controls. Shown in orange is the PHS-tolerant genotype Redhead (ID 300) and shown in yellow is the PHS-susceptible control, QQ74 (ID 9). Bars shown are standard error bars.

**Figure 4 plants-13-01297-f004:**
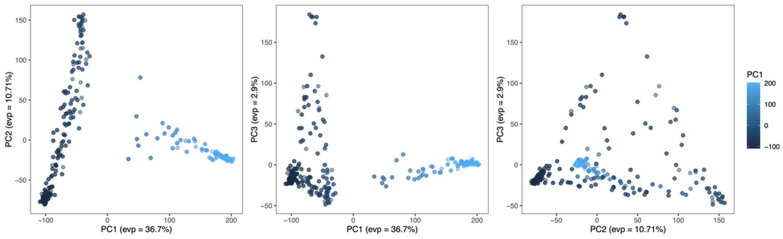
Principal Component Analysis for quinoa accession population with Explained Variance Percentage (EVP). Principal components were derived from 246 quinoa accessions based on 48,025 markers. Each dot represents an accession, with colors indicating their PC1 values. PC1 explained 36.7% of the phenotype variance, followed by PC2, which explains 10.71% of the variation. PC3 contributes 2.9% to the overall variance explanation. The population exhibits a division into two clusters based on positive or negative PC1 values.

**Figure 5 plants-13-01297-f005:**
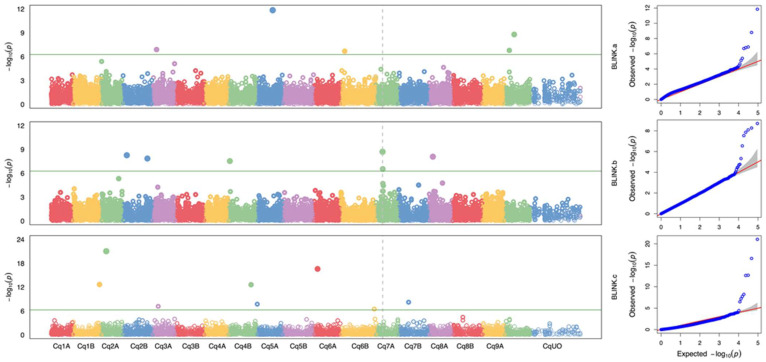
Manhattan and quantile–quantile (QQ) plots depicting genome-wide association analysis for preharvest sprouting (PHS). The plots were generated using BLINK, employing the first three principal components (PCs) and one indicator (based on PC1 value) as the coefficient of covariate. The PHS-associated markers for different sigmoid parameters a, b, and c are positioned above the Bonferroni line (green line), calculated at a 0.05 significance level with 48,025 markers. In the QQ plot, the gray shaded area denotes the 95% confidence interval under the null hypothesis.

**Figure 6 plants-13-01297-f006:**
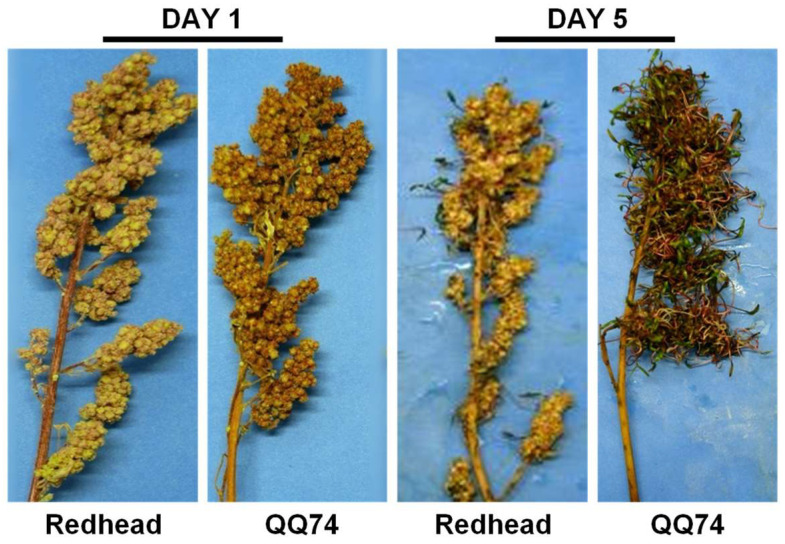
Quinoa controls used for PHS panicle screen. Differences in panicle sprouting between the PHS-tolerant (genotype Redhead) and PHS-susceptible (QQ74 genotype) controls with 5 days of continuous misting in a greenhouse chamber.

**Figure 7 plants-13-01297-f007:**
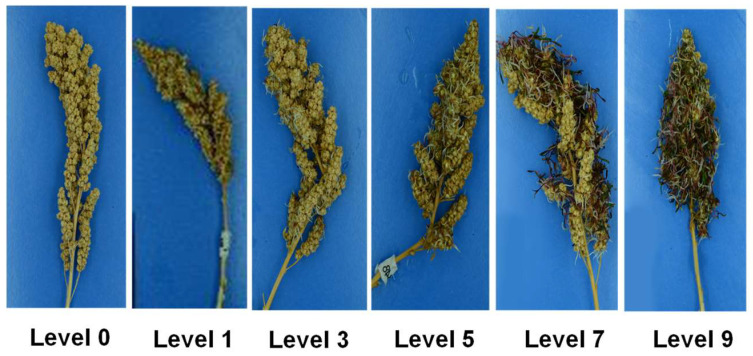
PHS scoring scale for quinoa panicles. Levels are based on the relative percentage of sprouted seeds in the quinoa panicle. Level 0 indicates no sprouting; Level 1 indicates the presence of radicle emergence (white) and a sprouting range from 1 to 19%; Level 3 indicates first observance of hypocotyls (pink-red) and a sprouting range from 20 to 39%; Level 5 is indicated by an intensification of the red color in the hypocotyls and a sprouting range from 40 to 59%; Level 7 is indicated by the observation of first true leaves and a sprouting range from 60 to 79%; Level 9 is indicated by the presence of mold and > 80% sprouting. Quinoa variety Redhead is pictured.

**Figure 8 plants-13-01297-f008:**
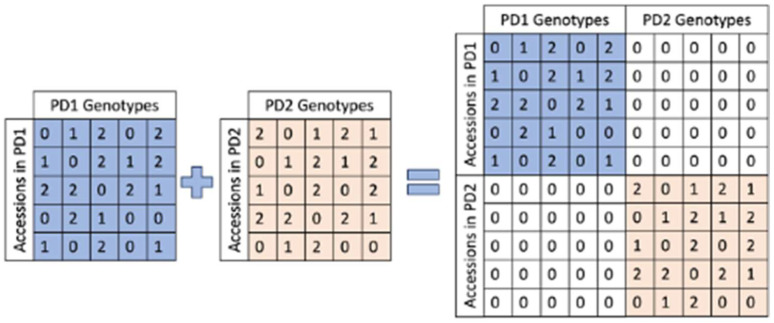
Construction of additive–additive model for quinoa accession genotypes. Quinoa accession genotypes are encoded as 0, 1, and 2, where 0 and 2 signify homozygous genotypes, and 1 represents heterozygous genotypes. Genotype matrices from planting date 1 (PD1) and planting date 2 (PD2) are concatenated for the downstream association study. The diagonals are filled with two zero matrices.

**Table 1 plants-13-01297-t001:** BLASTing against the *Chenopodium quinoa* genome revealed matching loci for seventeen SNP markers on the NCBI platform. Functional annotations for the putative genes were obtained from NIH DAVID (Database for Annotation, Visualization, and Integrated Discovery) bioinformatics website.

SNP	Locus	Gene List (DAVID)
Cq1B_66402375	XM_021904375	basic leucine zipper 23-like (*LOC110724880*)
Cq2A_11615394	XM_021877804	uncharacterized mitochondrial protein AtMg00810-like (*LOC110700265*)
Cq2B_60857052	XM_021883400	glutamyl-tRNA(Gln) amidotransferase subunit C, chloroplastic/mitochondrial-like (*LOC110705514*)
Cq2B_9387424	XM_021895300	putative calcium-transporting ATPase 11, plasma membrane-type (*LOC110716652*)
Cq3A_10231545	XM_021874234	transcription termination factor MTERF2, chloroplastic-like (*LOC110696907*)
Cq3A_14217269	XM_021883755	uncharacterized *LOC110705822* (*LOC110705822*)
Cq4B_4014100	XR_002507314	uncharacterized *LOC110701120* (*LOC110701120*)
Cq4B_56717106	XM_021880529	katanin p80 WD40 repeat-containing subunit B1 homolog (*LOC110702783*)
Cq5A_582429	XM_021887486	protein MAIN-LIKE 1-like (*LOC110709270*)
Cq6A_8709044	XM_021911740	uncharacterized *LOC110731845* (*LOC110731845*)
Cq6B_83520732	XM_021858518	ABC transporter G family member 11-like (*LOC110682235*)
Cq6B_9876457	XR_002509647	uncharacterized *LOC110712666* (*LOC110712666*)
Cq7A_17313829	XM_021919715	peter Pan-like protein (*LOC110739250*)
Cq7A_17874020	XM_021919734	heavy metal-associated isoprenylated plant protein 36-like (*LOC110739269*)
Cq8A_9702538	XM_021910407	pentatricopeptide repeat-containing protein At1g71490-like (*LOC110730586*)
Cq9B_12163476	XM_021894216	G-type lectin S-receptor-like serine/threonine-protein kinase At4g27290 (*LOC110715629*)
Cq9B_24343108	XR_002510671	uncharacterized *LOC110717110* (*LOC110717110*)

## Data Availability

The authors confirm that the data supporting the findings of this study are available within the article, Appendix A, and from the corresponding authors, (KMM and ALH), upon reasonable request.

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
