# Peer review of "Preharvest Sprouting in Quinoa: A New Screening Method Adapted to Panicles and GWAS Components"

_plants, 2024, doi:10.3390/plants13101297_

Round 1

Reviewer 1 Report

Comments and Suggestions for Authors

The draft by Ocaña-Gallegos et al. (2024), titled "Preharvest Sprouting in Quinoa:….", which was sent to Plants for possible publication, was assessed.

- Any Introduction should be clear and precise, ensuring that its content remains highly relevant and actual. The first paragraph of this, which refers to the seed pre-harvest sprouting (PHS), requires updated references. PHS has been thoroughly reviewed very recently by several renowned authors. The refs. 1, 3, and 7 are outdated and should be replaced with more current ones. The 2nd paragraph is written in a confusing manner and requires careful revision;  further, the refs. 9-11 are not properly written in the reference list, again indicating that the authors have not thoroughly reviewed the text. The third paragraph, referring to types of dormancy, is written in an unclear physiological way. It should be rewritten taking into account a large number of recent updates on the subject. Refs. 13 (2004), 14 (2005), 15 (2008), 16 (1976) and 17 (1961), are from 16-72 years ago, which is not acceptable for a 2024 manuscript. Similarly, I fail to see the scientific justification for including the fourth paragraph in this draft. Therefore it should be removed. The fifth paragraph is overly lengthy, and the reader is not adequately informed about the authors' intended message. It's important to remember that this Journal typically addresses physiological issues in plants. Finally, a clearer version of the last paragraph must be written. This paragraph, which is always pivotal in any draft, should clarify the main reasons for using GAPIT3 and BLINK. As an example: “These software were chosen for their strong ability in conducting GWAS and identifying SNPs related to PHS in quinoa. By utilizing these advanced tools, the study aims to uncover the genetic basis of PHS and identify potential candidate genes or genomic regions associated with this trait”. Together, given the numerous scientific weaknesses in the current Introduction, the authors should undertake an intense editing process to transform the text into one that is better written and acceptable.

            - The text of the legend of Fig. 1 is confusing.  

            - Final of p. 3: …..the spread of scores in planting date 2 (PD2) was larger than in planting date 1 (PD1), likely due to delays in harvesting PD2 attributable…..  

            - The text corresponding to Figs. 1 and 2, and S1 is very difficult to understand. Once again, the way the authors express themselves falls far short of scientific reality. Furthermore, the writing has not been carefully crafted at all.

            - 2.1. and 2.2. In this strange text, the Ref. is not sure whether is evaluating a text related to a physiological process or rather a text from the field of complex statistics and mathematics….. Do you want examples?: (i) Addressing Population Stratification…. an Indicator; (ii) a simple linear regression model; (iii) steepest slopes; (iv) PHS-susceptible control; (v) Pearson correlation; (vi) potential confounding effects; (vii) Coefficient of Variation (CV); (viii) association analysis (Figure S3A&B).

            - The use of the KEGG database and DAVID (i.e., functional annotation analysis) should be justified to relate it to the different tools employed in this draft. What benefit do the authors derive from the data in Table 1?

            - Regarding Discussion and ConclusionsIf this work done on quinoa provides several important benefits, the authors should cite and discuss them one by one and as a whole. In the conclusions, the authors only refer to prospects. All together, this work requires significant modification to clarify the reasons for its existence and to rationalize its writing so that readers, and more specifically in this case, the referees, can understand the reasons for its physiological approach.

Author Response

The authors thank the reviewer for the helpful feedback and comments. Where possible we have incorporated changes based on the reviewer’s suggestions in order to improve clarity and cohesiveness. The reviewer's comments are addressed sequentially and labeled with “RR" for response to review in blue text below.

The draft by Ocaña-Gallegos et al. (2024), titled "Preharvest Sprouting in Quinoa….", which was sent to Plants for possible publication, was assessed.

1-2.: Any Introduction should be clear and precise, ensuring that its content remains highly relevant and actual. The first paragraph of this, which refers to the seed pre-harvest sprouting (PHS), requires updated references. PHS has been thoroughly reviewed very recently by several renowned authors. The refs. 1, 3, and 7 are outdated and should be replaced with more current ones. The 2nd paragraph is written in a confusing manner and requires careful revision;  further, the refs. 9-11 are not properly written in the reference list, again indicating that the authors have not thoroughly reviewed the text. 

RR1-2: Additional citations have been added, and citation formatting has been corrected.

  1. The third paragraph, referring to types of dormancy, is written in an unclear physiological way. It should be rewritten taking into account a large number of recent updates on the subject.

RR3. The authors have revised this paragraph for clarity and added additional citations.

  1. Refs. 13 (2004), 14 (2005), 15 (2008), 16 (1976) and 17 (1961), are from 16-72 years ago, which is not acceptable for a 2024 manuscript.

RR4. The authors thank the reviewer for the suggestions and have added newer citations for the identified references. The authors also identified that reference 17 was not accurately cited and has been corrected. Additionally, references 16 and 17 are the original methods papers for the FN and spike wetting methods. Neither of these methods have undergone changes since first report and remain the standard. Therefore, the authors intend to leave them in. However, we have added the FGIS Directive for the Falling numbers method.

  1. Similarly, I fail to see the scientific justification for including the fourth paragraph in this draft. Therefore it should be removed. The fifth paragraph is overly lengthy, and the reader is not adequately informed about the authors' intended message. It's important to remember that this Journal typically addresses physiological issues in plants.

RR5. The authors appreciate the feedback about this paragraph. However, the fourth paragraph is important because we are talking about other possible ways in which PHS could be assessed in quinoa and giving justification of why we are choosing spike wettings tests over FN or alpha-amylase enzyme tests. The language has been changed to make this point clear.  Paragraph 5 has also been broken into two which the authors hope adds additional clarity.

  1. Finally, a clearer version of the last paragraph must be written. This paragraph, which is always pivotal in any draft, should clarify the main reasons for using GAPIT3 and BLINK. As an example: “These software were chosen for their strong ability in conducting GWAS and identifying SNPs related to PHS in quinoa. By utilizing these advanced tools, the study aims to uncover the genetic basis of PHS and identify potential candidate genes or genomic regions associated with this trait”. Together, given the numerous scientific weaknesses in the current Introduction, the authors should undertake an intense editing process to transform the text into one that is better written and acceptable.

RR6. The authors appreciate the reviewer’s suggestion to include the reason for selecting GAPIT 3 and BLINK over other options. The rationale has been incorporated as suggested.

  1. The text of the legend of Fig. 1 is confusing.  

RR7. Text has been revised for clarity.

  1. Final of p. 3: …..the spread of scores in planting date 2 (PD2) was larger than in planting date 1 (PD1), likely due to delays in harvesting PD2 attributable…..  

RR8. Description has been added. Due to formatting some of the information seems to have been lost, but has been added back.

  1. The text corresponding to Figs. 1 and 2, and S1 is very difficult to understand. Once again, the way the authors express themselves falls far short of scientific reality. Furthermore, the writing has not been carefully crafted at all.

RR9. As per the suggestion above, the legend for Fig 1 has been edited. However, the authors respectfully disagree with the assessment for the captions in Figure 2 and S1 request these be kept in as written. If the reviewer requests additional revision, the authors would ask for more specificity about how.

  1. 2.1. and 2.2. In this strange text, the Ref. is not sure whether is evaluating a text related to a physiological process or rather a text from the field of complex statistics and mathematics….. Do you want examples?: (i) Addressing Population Stratification…. an Indicator; (ii) a simple linear regression model; (iii) steepest slopes; (iv) PHS-susceptible control; (v) Pearson correlation; (vi) potential confounding effects; (vii) Coefficient of Variation (CV); (viii) association analysis (Figure S3A&B).

RR10. Indeed, in these two sections of the results is where our biological findings are explained in mathematical terms. Section 2.1 explains the creation of a ranking from the least sprouting quinoa genotype, to the most sprouting one. This inevitably involves using terms such as (ii) a simple linear regression model; (iii) steepest slopes; (iv) PHS-susceptible control and (v) Pearson correlation. Section 2.2 explains how to adequately choose a PC that will be included in the GWAS study. We apologize if the referees were expecting answers in terms of physiological processes, but that was not the main aim of our paper. Our study aimed at: 1) adapting a PHS screening method to panicles, 2) assess a diverse panel of quinoa genotypes to determine which are susceptible or resistant to PHS, and 3) to perform GWAS and propose potential genes underpinning the PHS trait in quinoa. The terminology used to describe the analysis is consistent with other studies using the same methodology for analysis.

  1. The use of the KEGG database and DAVID (i.e., functional annotation analysis) should be justified to relate it to the different tools employed in this draft. What benefit do the authors derive from the data in Table 1?

RR11. The authors appreciate the reviewers for bringing this to our attention. The use of functional annotation with bioinformatic software and methos was essential in our study. Given quinoa’s status as a new crop with limited annotation information for downstream GWAS, we employed bioinformatic methods to predict the potential function of the PHS-associated markers. These predicted functions could offer valuable insights for downstream research. We have made the clarification in the discussions.

  1. Regarding Discussion and Conclusions:  If this work done on quinoa provides several important benefits, the authors should cite and discuss them one by one and as a whole. In the conclusions, the authors only refer to prospects.

RR12. After reviewing the Discussion and Conclusion sections that authors respectfully request to leave each section as written. Within the Discussion section we have provided commentary on the results and also next steps as a typical Discussion/Conclusions section. We have also used the Conclusions section to discuss future implications of the work beyond the scope of the immediate results. The Conclusion section in Plants is “not mandatory but can be added to the manuscript if the discussion is unusually long or complex.”  While we do not feel out Discussion section is unusually long or complex, we also think it is our use of the “Conclusion” was appropriate.

Reviewer 2 Report

Comments and Suggestions for Authors

Review is on the attached file.

Author Response

Please see the attachment for response to review.

The authors thank the reviewer for the helpful feedback and comments. Where possible we have incorporated changes based on the reviewer’s suggestions in order to improve clarity and cohesiveness. The reviewer's comments are addressed sequentially and labeled with “RR" for response to review in blue text below.

  1. The title could be improved. “GWAS components” do not sound accurate.

RR1. Given that GWAS stands for Genome-wide association study, it’d be inaccurate or redundant to describe it as GWAS study, or GWAS analysis. Plus, it was analyzed through the manipulation of 3 components: a - the upper asymptote, b -growth rate, c - time of maximum growth. Therefore, we would like to leave it as is.

2.Abstract: PNW lacks definition before abbreviation.

RR2. Thank you for the feedback. This has been corrected.

  1. Key words must be different from the title.

RR3. The authors have edited this.

  1. Introduction: Use scientific name of quinoa in the first time it appears.

RR4. Thank you, the scientific name has been added.

  1. “if quinoa is the same as other seeds” (page 2) – This sentence is rather subjective. Please re-write.

RR5. We have changed the wording to be less subjective and more specific on page 2.

  1. “Relatives of quinoa” – There was no characterization of quinoa scientific name to know if these species are really relatives.

RR6. Quinoa’s scientific name has been included in the introduction.

  1. “World Core Collection (WCC) – Where is this WCC? In what Institution?

RR7. This information has been added, and is found in the ‘Germplasm’ and ‘Acknowledgements’.

  1. “If the nature of PHS in quinoa resembles that of cereals” (page 3) – Again, this is too subjective for a scientific paper.

RR8. We thank the reviewer for this suggestion. The language had been modified for clarity on page 3.

  1. PD2 and PD1 (final paragraph in page 3) lack definition.

RR9. Definition has been added.

  1. Figure 2: two dots in the final sentence.

RR10. This has been corrected.

  1. The amount of variance explained by the three first principal components is too low (40%). I strongly recommend excluding the PC2 x PC3 plot in Figure 4 as it lost the two-cluster separation.

RR11. The authors appreciate the reviewer's suggestions. Upon reevaluation, it was found that the overall explained variance percentage (EVP) for the first three principal components (PCs) is 50.31%, rather than 40% as previously stated. This correction has been reflected in both the PC plot and its accompanying caption. Additionally, detailed EVP values for each PC have been provided. The PCA results highlight the significance of PC1 in capturing the major sources of phenotype variance (36.7%), followed by PC2 (10.71%) and PC3 (2.9%), all of which contribute to the overall variance explanation. The subplot depicting PC1 vs PC2 and PC1 vs PC3 demonstrates clear population structures delineated by PC1 values. Conversely, the PC2 vs PC3 plot does not exhibit such separation. This comparison elucidates why PC1 values were utilized as indicators and included in the model as covariates. Therefore, we maintain the inclusion of the PC2 vs PC3 plot in the analysis.

  1. Table 1: Loci should come in italics.

RR12. This has been reformatted.

  1. Authors could have explored quinoa domestication in PHS tolerance.

RR13. The authors appreciate this perspective and have added a statement about future work investigating the implications of domestication of PHS tolerance in quinoa. Given that the main point of this manuscript is about the development of a new method, the authors decided to be more conservative in scope.

  1. What is PD? It should be defined prior its first use (page 7).

RR14. The definition of PD is now described on page 3.

  1. Material and methods: WSU – Please define.

RR15. Thank you for this catch. Washington State University has now been added.

  1. Table S2 provides slope values. It should be Table S3 (page 8 - Germplasm and Experimental design and greenhouse conditions).

RR16. Thank you, this has been changed

  1. Figure 5: There are two dots at the end of the sentence.

RR17. We have made this change.

  1. Development of PHS scoring scale: Figure S2 are the Manhattan plots. It should be Figure S3.

RR18. This has been fixed.

  1. Heading 4.4 is repeated in relation to heading 4.3.

RR19. Thank you, this has been changed to ´Data collection & analysis´

  1. Figure 7 (page 11): This figure should come in results.

RR20. We strongly agree with the reviewer’s suggestion and have relocated Figure 7 to the Results section.

  1. Authors should explain why they used Blink method instead of Farm CPU.

RR21. In the Blink citation, they have demonstrated through both real and simulated data analyses that Blink offers improved statistical power compared to FarmCPU, while also significantly reducing computing time. The rationale for choosing Blink over FarmCPU has been provided in the last paragraph of the introduction section.

  1. Page 12 “If quinoa PHS....alpha-amylase activity”- This is not a conclusion of your work. It is rather subjective.

RR22. Thanks for bringing this point to our attention, we agree it is better not hypothesize about future research results. This statement has been removed

Reviewer 3 Report

Comments and Suggestions for Authors

Preharvest sprouting (PHS) is undesirable sprouting of seeds that occurs before harvest and is triggered by rain or humid conditions and is responsible for yield losses and lower nutrition in cereal grains. PHS has been extensively studied in wheat, barley, and rice, but there are limited reports for quinoa. This manuscript aimed to better understand PHS in quinoa by adapting a PHS screening method commonly used in cereals. This involved carrying out panicle-wetting tests and developing a scoring scale specific for panicles to quantify sprouting. The findings from this study indicate that PHS occurs at varying degrees across a subset of the quinoa germplasm tested and that it is possible to access PHS tolerance from natural sources. Ultimately, these genotypes can be used as parental lines in future breeding programs aiming to incorporate tolerance to PHS. In general, this manuscript is well written, the experimental design was well organized and data analyses were good and produced some interesting results, which will use for tolerance to PHS for quinoa. I have only the following minor comments:

Comments:

1.     The first letter of key words should be capital;

2.     Figure 1, the labeling words  and digital data numbers are too small, should make them bigger ;

3.     For the discussion part, it should add more comparison with previous studied in the other species;

4.     Literature need to be updated, there is hardly any references from 2023-2024;

5.     References are not consistent, some have DOI number and some not, all the references should be in the same pattern.

Author Response

Please see the attached response to review. 

The authors thank the reviewer for the helpful feedback and comments. Where possible we have incorporated changes based on the reviewer’s suggestions in order to improve clarity and cohesiveness. The reviewer's comments are addressed sequentially and labeled with “RR" for response to review in blue text below.

  1. The first letter of key words should be capital;

RR1. Thank you, this correction has been made.

  1. Figure 1,the labeling words and digital data numbers are too small, should make them bigger ;

RR2. We believe this might have to do with formatting at the editorial end and hope they will show all our figures at the correct size.

  1. For the discussion part, it should add more comparison with previous studied in the other species;

RR3: The authors respect the reviewer’s comment. The authors do mention the results from our study are different than those previously reported for PHS wheat. Given the limited or absence of information for other plants, the authors felt that trying to make comparisons to other plants was beyond scope and not appropriate.

  1. Literature need to be updated, there is hardly any references from 2023-2024;

RR4: The authors have added newer citations where appropriate throughout the manuscript.

  1. References are not consistent, some have DOI number and some not, all the references should be in the same pattern.

RR5. All of the citations have been revised for consistency.
